# Clinical and Genetic Features of Korean Patients with Achromatopsia

**DOI:** 10.3390/genes14020519

**Published:** 2023-02-18

**Authors:** Yong Je Choi, Kwangsic Joo, Hyun Taek Lim, Sung Soo Kim, Jinu Han, Se Joon Woo

**Affiliations:** 1Department of Ophthalmology, Seoul National University College of Medicine, Seoul National University Bundang Hospital, Seongnam 13620, Republic of Korea; 2Department of Ophthalmology, University of Ulsan College of Medicine, Asan Medical Center, Seoul 05505, Republic of Korea; 3Orthopia Eye Clinic, Seoul 06162, Republic of Korea; 4Institute of Vision Research, Department of Ophthalmology, Severance Hospital, Yonsei University College of Medicine, Seoul 03722, Republic of Korea; 5Institute of Vision Research, Department of Ophthalmology, Gangnam Severance Hospital, Yonsei University College of Medicine, Seoul 06273, Republic of Korea

**Keywords:** achromatopsia, *CNGA3*, *CNGB3*, *PDE6C*, *GNAT2*, Korean population

## Abstract

This multicenter study aimed to characterize Korean patients with achromatopsia. The patients’ genotypes and phenotypes were retrospectively evaluated. Twenty-one patients (with a mean age at the baseline of 10.9 years) were enrolled and followed up for a mean of 7.3 years. A targeted gene panel or exome sequencing was performed. The pathogenic variants of the four genes and their frequencies were identified. *CNGA3* and *PDE6C* were equally the most prevalent genes: *CNGA3* (N = 8, 38.1%), *PDE6C* (N = 8, 38.1%), *CNGB3* (N = 3, 14.3%), and *GNAT2* (N = 2, 9.5%). The degree of functional and structural defects varied among the patients. The patients’ age exhibited no significant correlation with structural defects. During the follow-up, the visual acuity and retinal thickness did not change significantly. In *CNGA3*-achromatopsia patients, a proportion of patients with a normal foveal ellipsoid zone on the OCT was significantly higher than that of patients with other causative genes (62.5% vs. 16.7%; *p* = 0.023). In *PDE6C*-achromatopsia patients, the same proportion was significantly lower than that of patients with other causative genes (0% vs. 58.3%; *p* = 0.003). Korean patients with achromatopsia showed similar clinical features but a higher prevalence of *PDE6C* variants than those of other ethnic groups. The retinal phenotypes of the *PDE6C* variants were more likely to be worse than those of other genes.

## 1. Introduction

Achromatopsia (ACHM) is a rare genetic disease with an autosomal recessive (AR) inheritance. It is characterized by the cone-specific loss of the retinal anatomy and function. The disease presents with a decrease in daylight vision, which begins early in life. Its diagnosis is based on clinical presentations and genetic tests to identify variants in the known causative genes. Approximately 80–90% of cases have causative variants in *CNGA3* and *CNGB3* [1,2]. Others have variants in *PDE6C*, *GNAT2*, *ATF6*, and *PDE6H* [2]. However, the prevalence of each gene differs greatly according to region and ethnicity [2].

Because of its cone-specific loss-of-function, ACHM is considered a good candidate for gene replacement therapy. The severity and extent of photoreceptor (PR) damage are important factors in gene replacement therapy [3]. Many studies have depicted the genotypes and clinical presentations of ACHM in several populations [1,4,5,6,7] and have demonstrated methods to evaluate the remnant PR in patients with ACHM [1,8,9,10,11].

The clinical decisions for the timing of gene replacement therapy are influenced by the remodeling stages of the retina [3]. The retinal tissue is post-mitotic so that it is non-regenerating after irreversible damage occurs. Therefore, the natural history of the disease must be considered to not miss the optimal window for treatment. Although ACHM was traditionally categorized as a stationary cone dysfunction [12], many studies have reported evidence of its progression [4,9,10].

The genotypes and phenotypes of the variants in ACHM patients in Korea are not well known and are both important for the future implementation of gene therapy. We identified the genotypes, phenotypes, and their correlations in Korean patients with ACHM. Longitudinal changes in the retinal structure and function were also analyzed to elucidate the possible progression and its relationship to the genotype.

## 2. Materials and Methods

### 2.1. Patient Selection

We retrospectively reviewed the medical records of patients diagnosed with ACHM. Patients who visited two tertiary referral hospitals between January 2007 and December 2020 were enrolled. The diagnosis was based on genetic testing and clinical presentations, such as the early onset of poor visual acuity, infantile nystagmus, color vision defects, and abnormal cone responses with normal or subnormal rod responses on an electroretinogram (ERG) [13]. We first reviewed the records of patients who underwent genetic testing for causative variants in 6 known causative genes of ACHM. The clinical presentations were then reviewed to exclude patients who had clinical features that did not correspond to those of ACHM.

### 2.2. Ophalmic Examinations

Ophthalmic examinations, including the measurement of the best-corrected visual acuity (BCVA), refractive errors, a biomicroscopy with a slit-lamp microscope, and fundus examinations, were performed. Ancillary tests were conducted, including color fundus photography (VX-10a; KOWA, Tokyo, Japan), optical coherence tomography (OCT; SPECTRALIS HRA+OCT, Heidelberg Engineering, Heidelberg, Germany), and an ERG (1) (VERIS, Electro-Diagnostic Imaging, Milpitas, CA, USA) or (2) (Ret-eval, LKC, Gaithersburg, MD, USA). The ERG was recorded according to the International Guidelines of the International Society of Clinical Electrophysiology of Vision (ISCEV) [14,15]. Visual acuity and central retinal thickness (CRT) were compared between the baseline and last measurements. Visual acuity was converted to the logarithm of the minimum angle of resolution (LogMAR). Statistical tests were conducted on changes in the BCVA and CRT and the rates of the changes. The denominator of the rate of change was the period between the first and the last observations. Color vision tests were performed in some cases using an Ishihara pseudoisochromatic plate test or a Hardy–Rand–Rittler (HRR) pseudoisochromatic plate test.

Evaluation of rod functions in the scotopic ERG was based on the amplitude of the b-wave in DA 0.01 ERG. Cone functions were analyzed based on the amplitudes of a- and b-waves in an LA 3.0 ERG or the N1-P1 amplitude of a 30 Hz flicker ERG. The reference values were provided by the ERG equipment that was based on the accumulated normative data.

The existence of one or more inner layers in the fovea was noted as foveal hypoplasia, as described in the existing literature [1]. The severity of degeneration shown on the OCT was graded in four categories. Grade 1 was considered normal. Grade 2 was defined as disruptions in the photoreceptor layers, particularly in the inner segment of the ellipsoid (ISe) zones. Grade 3 was defined as the hyporeflective zone in the ISe. Grade 4 was defined as the atrophy of the outer retinal layer and retinal pigment epithelium. The same category has been described in the existing literature [1], except that we did not adopt a stage denoting the absence of the ISe.

### 2.3. Genotyping

Patient genotyping was performed using targeted gene panels or whole-exon sequencing. A customized target enrichment kit (Celemics, Seoul, Republic of Korea) was designed, covering the exon and splicing regions of 254 genes related to inherited retinal diseases. Segregation was performed when possible. Sequencing and bioinformatics analyses were performed according to processes described in the existing literature [16]. The pathogenicity of the variants was evaluated based on guidelines suggested by the American College of Medical Genetics (ACMG) [17]. Accession numbers of the transcripts were NM_001298.2 for *CNGA3*, NM_019098.4 for *CNGB3*, NM_006204.3 for *PDE6C*, and NM_005272.3 for *GNAT2*. To detect the deletion of exons, we used read depth-based detection of the copy number variations (CNVs). Analysis and visualization of CNV were conducted using Celemics CNV software (Celemics, Seoul, Republic of Korea) or ExomeDepth version 1.1.10 [18], after using a base-level read depth normalization algorithm designed by the authors [19].

### 2.4. Statistical Analysis

All statistical analyses were performed using Python (Python Language Reference, version 3.10.4) and using Pandas and SciPy libraries. Data were expressed as numbers, percentages, and the mean ± standard deviation. A one-tailed or two-tailed Boschloo’s exact test for non-numerical data was conducted to compare categorical variables. The *p*-value was compensated using the Bonferroni correction in multiple pairwise comparisons. A Mann–Whitney U test was used to compare continuous variables between the two groups based on the results of the normality tests. To compare paired samples of continuous variables, a Wilcoxon signed-rank test was used. A 95% confidence interval and 5% significance level were adopted. Results were considered statistically significant when a *p*-value of less than 0.05 was observed.

## 3. Results

### 3.1. Demographics and Clinical Presentation

Twenty-one patients with ACHM were enrolled. The demographic and clinical features are shown in Table 1. Thirteen patients were men, and eight patients were women. The mean age at the initial visit was 10.9 ± 12.9 (0.5–47.2) years. The patients were followed up for a mean of 7.3 ± 6.8 (0–25) years. The LogMAR-converted mean BCVAs at the initial visit were 1.01 ± 0.28 (0.52–1.52) in the right and 1.02 ± 0.26 (0.52–1.52) in the left eyes. The LogMAR-converted mean BCVAs at the last visit were 0.98 ± 0.2 (0.7–1.4) in the right and 0.97 ± 0.2 (0.7–1.4) in the left eyes. The mean changes of the LogMAR-converted BCVA were −0.05 ± 0.21 (−0.57–0.30) in the right and −0.05 ± 0.20 (−0.4–0.48) in the left eyes. In 18 patients, the BCVA was measured longitudinally. The mean follow-up duration was 7.4 ± 6.2 years. As shown in Figure 1, no significant changes in the BCVA were observed in either eye of those patients (right: *p* = 0.385; left: *p* = 0.239). Normalized to the follow-up duration, the changes in the BCVA per year were −0.017 ± 0.056 (−0.185–0.059) in the right and −0.02 ± 0.046 (−0.128–0.042) in the left eyes, which were not significant (right, *p* = 0.285; left, *p* = 0.114). All patients had infantile nystagmus, but the nystagmus in one patient (#13) had resolved spontaneously during early adolescence. The median refractive errors were −3.01 ± 6.42 (−14.25–7.00) diopters in the right and −2.89 ± 6.19 (−14.25–7.00) diopters in the left eyes. Twelve patients underwent color vision testing, which resulted in total color blindness. One patient (#21) failed to undergo the HRR pseudoisochromatic plate test. Another patient (#19) refused all ophthalmic examinations and ancillary tests except for genetic testing and OCT.

### 3.2. Retinal Phenotypes

Varying degrees of foveal abnormalities were observed. The representative fundus photographs and OCT images are shown in Figure 2. Seventeen patients (85%) revealed no abnormalities in the fundus photographs. One (5%, #4) patient had hypopigmentary changes of the retinal pigment epithelium (RPE); the other two (10%, #13, #16) had atrophy of the retina and RPE. OCT revealed a wide range of degeneration in the foveal photoreceptors (Table 1). At the latest visit to each patient, seven (35%) patients had a continuous ISe. Six patients (30%) experienced ISe disruptions. Four patients (20%) had hyporeflective zones. Three patients (15%) exhibited atrophy of the outer retina and the RPE. All patients had symmetry in both eyes on the OCT, except one (#4, right: ISe disruption; left: normal). One toddler (#10) did not cooperate with the OCT during the entire follow-up period. Foveal hypoplasia was identified in six patients on OCT (30%).

The measurement of the CRT was performed longitudinally for 13 patients. The mean follow-up duration was 5.0 ± 2.8 (1.1–11.9) years. At the first visit of each patient, when the OCT was conducted, the mean CRT was 187.8 ± 46.2 (102–242) μm in the right eyes and 186.9 ± 48.8 (101–243) μm in the left eyes. At the latest visit of each patient, when the OCT was conducted, the mean CRT was 186.8 ± 46.5 (101–238) μm in the right eyes and 182.5 ± 47.9 (102–238) μm in the left eyes. Figure 1B shows the changes in the CRT for each patient. There were significant changes in the CRT in the left eye (*p* = 0.04). However, the CRT in the right eye did not change (*p* = 0.329). After normalizing to the interval between the measurements, the mean rate of change was −0.5 ± 1.1 (−2.7–1.1) μm/year in the right eyes and −1.0 ± 1.7 (−4.9–1.0) μm/year in the left eyes. The mean rate of change was significant in the left eye (*p* = 0.02) but not in the right eye (*p* = 0.057).

### 3.3. Electroretinogram

The ERG was conducted in 17 cases (Table 2). The representative images of the ERG are shown in Figure 3. The dark-adapted (DA) 0.01 ERG was observable in all of the patients. The mean amplitude of the b-wave in the DA 0.01 ERG was 69.7 ± 63.3 (7.9–215) μV in the right eyes and 69.5 ± 56.3 (8.3–196) μV in the left eyes. In 16 patients (94%), the light-adapted (LA) 3.0 ERG showed no measurable amplitudes of the a- and b-waves. In the flicker ERG, sixteen patients (94%) showed no measurable responses.

### 3.4. Genotypes

Biallelic variants were also identified in 21 patients. They had pathogenic variants in four common causative genes of ACHM. The causes of ACHM were defects in *CNGA3* in eight patients (38.1%), *CNGB3* in three patients (14.3%), *PDE6C* in eight patients (38.1%), and *GNAT2* in two patients (9.5%) (Table 3). Ten patients were compound heterozygotes, and three were homozygotes. Since the segregation was not conducted in the other eight patients who were heterozygotes, it was unidentified whether they were compound heterozygotes. Eleven of these variants were novel. The predicted pathogenicity of nine novel variants was “Likely pathogenic” or “Pathogenic,” according to the ACMG classification. The other two novel variants were predicted to be of “Uncertain significance.” In the patients with *CNGA3*-ACHM, eleven variants were missenses (79%), two variants were frameshifts (14%), and the other variant was an initiation codon variant (7%). Two variants in *CNGA3* were found in multiple patients: c.553C>G in patients #3 and #4 and c.1001C>T in patients #1 and #2. *CNGA3*:c.829C>T was detected in the homozygote. In patients with *CNGB3*-ACHM, three variants were splice-site variants (60%), one variant was a frameshift (20%), and the other was the deletion of exon 16 (20%). *CNGB3*:c.1928+2T>C was identified in a homozygote. In patients with *PDE6C*-ACHM, eleven variants (85%) were missense, one variant (8%) was a frameshift, and the other one (8%) was a deletion of exon 1. Two variants in *PDE6C*-ACHM were found in multiple patients: c.1771G>A in patients #12 and #16 and c.85C>T in patients #14 and #16. There were no homozygotes in patients with *PDE6C*-ACHM. In patients with variants in *GNAT2*, c.481C>T was found in two patients, #20 and #21 (67%). One patient was homozygous; the other had a frameshift variant (33%).

### 3.5. Correlations of Genotypes and Phenotypes

The correlation between the presence of ISe abnormalities and clinical features was evaluated (Table 4). Patients were stratified into four groups depending on their causative gene variants. Age (*p* = 0.43), BCVA (right: *p* = 0.443; left: *p* = 0.676), funduscopic findings (*p* = 0.535), and proportion of hypoplasia (*p* = 0.386) were not significantly different between the two groups of patients with *CNGA3*-ACHM and *PDE6C*-ACHM, respectively. Based on OCT findings, proportions of patients with ISe changes were denoted for each mutated gene. There was a significant difference in the proportion of patients with ISe changes between the groups of patients with *CNGA3*-ACHM and *PDE6C*-ACHM (*p* = 0.011) (Table 4). In patients with *CNGA3*-ACHM, the proportion of patients with no ISe change was significantly higher than that of the patients with variants in other causative genes (62.5% vs. 16.7%; *p* = 0.023) (Appendix A). In patients with *PDE6C*-ACHM, the proportion of patients with ISe changes was different from that of the patients with causative variants in other genes (100% vs. 58.3%; *p* = 0.003). Multiple pairwise comparisons with adjustment were conducted between groups of patients with causative variants in each mutated gene (Appendix A). The proportion of patients without ISe changes was higher in the group of patients with *CNGA3*-ACHM than that in the group of patients with *PDE6C*-ACHM (*p* = 0.005). The age of the patients was compared between the two groups that were separated by the existence of ISe changes in the OCT images, regardless of the causative genes. The mean ages of the groups of normal ISe and abnormal ISe were 12.5 ± 5.7 (6.4–23.9) and 22.0 ± 12.1 (7.6–47.2), respectively. The patients with a normal ISe were, on average, 9.5 years younger. However, the age did not differ significantly between the groups (*p* = 0.056). When compared to other causative genes, there were no significant differences in the proportion of patients with ISe changes in the groups of patients with *CNGB3*-ACHM and *GNAT2*-ACHM (Appendix A).

## 4. Discussion

In the present study, we analyzed the genotypes and phenotypes of 21 Korean patients with ACHM. To our knowledge, this is the first study of the genotypes and phenotypes in Korean patients. This was required, given the rapid and active development of gene therapy for ACHM.

We found a high proportion of *PDE6C* variants in Korean patients with ACHM. The genotyping of Korean patients has identified variants in *CNGA3*, *CNGB3*, *PDE6C*, and *GNAT2*. The frequencies of *CNGA3* and *PDE6C* were higher than those of other variants. Globally, the profiles of mutated genes vary widely in ACHM patients (Table 5) [1,4,7,30]. Koreans and Chinese are similar in the sense that they are East Asians. However, *CNGA3* is the most prevalently mutated gene in Chinese patients with ACHM [7]. Among them, only 0.8% of patients have variants in *PDE6C* [7]. Although there was a large difference, we considered it acceptable. Hanany et al. [34] reported the predicted genetic prevalence of genes in patients with AR-inherited retinal diseases based on the reported cases in public databases. According to them, the predicted number of ACHM patients with biallelic variants in *PDE6C* was 4418, which was 21.5% of all patients with ACHM in East Asia. This implies that more than 21.5% of the patients with ACHM had *PDE6C* variants in East Asia, especially in nations other than China. This indicates a potential market for investigation in East Asia for the gene therapy of *PDE6C*-ACHM.

We compared the OCT stages in four groups of patients with different causative genes for ACHM. OCT revealed that all eight patients with *PDE6C* variants had varying degrees of abnormalities in the ISe. The mean age of the patients with *PDE6C* variants was 13.2 years, which was lower than that reported by Georgiou et al. [9]. In the *PDE6C* variants, this high tendency to have severe retinal phenotypes was significantly different from that in patients with other causative variants: *CNGA3*, *CNGB3*, and *GNAT2* (Appendix A). In contrast, the proportion of normal OCT findings was significantly higher in the *CNGA3* variants (Appendix A). The mean ages of the patients with *CNGA3*-ACHM and *PDE6C*-ACHM were not significantly different (Table 4). However, the overall OCT grades of the patients with *PDE6C*-ACHM were significantly higher than those of the patients with *CNGA3*-ACHM (Appendix A). Thus, we expected a wide variation in the therapeutic window for gene therapy in Korean patients, given that the most common variants were CNGA3 and PDE6C.

In this study, a small proportion of patients had variants in *CNGB3* and *GNAT2*. However, the genetic profiles of the *CNGB3* variants coincided with those previously reported [27]. The variants in three patients with *CNGB3*-ACHM were nonsense, frameshift, and deletion of the exon variant. The loss of function in *CNGB3* is associated with complete ACHM that results in a complete loss of cone responses, a VA lower than 20/200, and a complete loss of color vision [36]. Among the 12 patients who did not undergo the color vision test, we could not identify whether the ACHM was a complete type. Coincidentally, none of the three patients with *CNGB3* variants underwent color vision testing. However, all had infantile nystagmus and optical vacancies in the layers of the outer retina on the OCT. Moreover, the mean LogMAR BCVAs were 1.12 ± 0.26 (0.82–1.43) in the right eye and 1.10 ± 0.17 (1.00–1.43) in the left eye at the last follow-up. These findings are relevant to the features of complete ACHM.

In addition to the lack of color vision tests in some cases, the small sample size is another limitation of this study. Because of the rare incidence of ACHM, it was unable to recruit a large number of patients, making the data prone to bias. Therefore, these limitations should be considered when interpreting the results of this study. Another limitation of this study is the lack of in-depth phenotyping of the retinal structure. Georgiou et al. [9] reported the usefulness of adapted optical scanning light ophthalmoscopy (AOSLO) for the in-depth phenotyping of ACHM. They revealed remnant PRs using AOSLO in a patient with a hyporeflective space at the fovea on the OCT. We revealed the fovea in four patients (22%) with hyporeflective spaces on the OCT images. They may have remnant PRs in the fovea that are more preserved than those shown in the OCT images. To select therapeutic targets in the future, more in-depth phenotyping using devices such as AOSLO will be necessary to avoid missing the therapeutic window.

## 5. Conclusions

To our knowledge, this was the first study to conduct genotyping, phenotyping, and analysis of their correlation in Korean patients with ACHM. We discussed the characteristic genetic prevalence and clinical presentation of ACHM in Korean patients. These findings should be considered for the future implementation of gene therapy in Korea.

## Figures and Tables

**Figure 1 genes-14-00519-f001:**
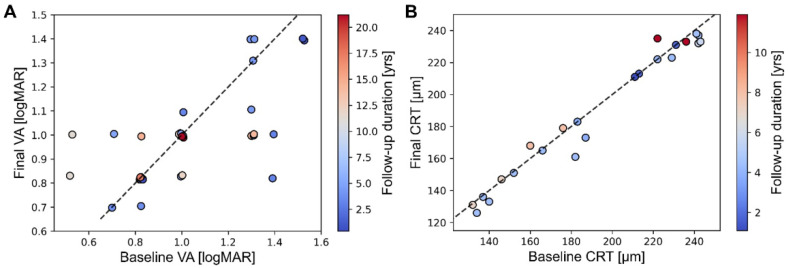
Changes in the best corrected visual acuity (BCVA) and central retinal thickness (CRT) in the patients with achromatopsia. The color of each mark denoted the follow-up duration between the baseline and the final visit when the BCVA or CRT was measured. (**A**). No significant changes in the BCVA were observed in either of the eyes. (**B**). A significant change in the mean CRT was observed in the left eyes, while it was not observed in the right eyes.

**Figure 2 genes-14-00519-f002:**
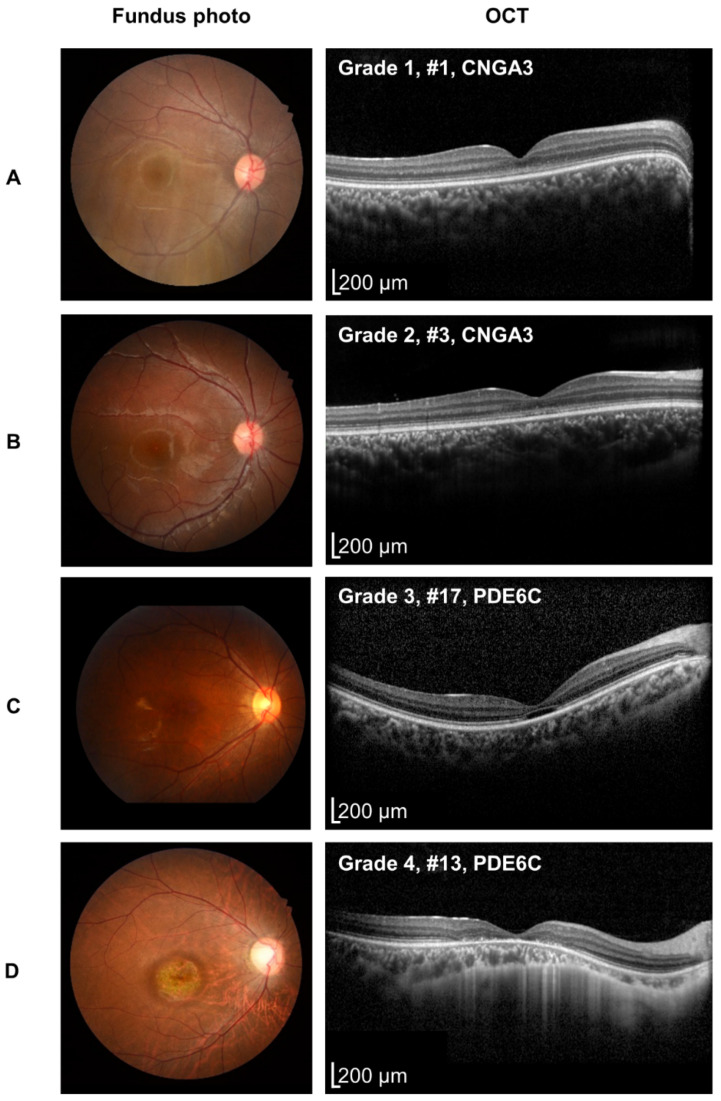
Representative images of multimodal ophthalmic imaging of patients showing various degrees of photoreceptor loss. Fundus photography and optical coherence tomography (OCT) were conducted. In the OCT, the degrees of photoreceptor loss varied into four categories; (**A**) continuous photoreceptor inner segment ellipsoid (ISe), (**B**) disruption of the ISe, (**C**) presence of hyporeflective zone, and (**D**) atrophy of the outer retina and retinal pigment epithelium.

**Figure 3 genes-14-00519-f003:**
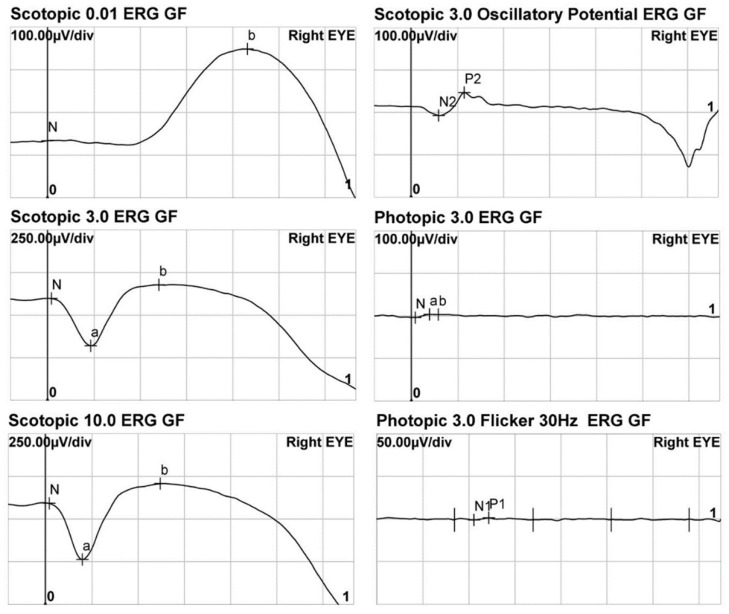
A representative result of the electroretinogram (ERG) obtained from patient #1. While almost no photopic responses were observed, scotopic responses were normal.

**Table 1 genes-14-00519-t001:** Clinical features and patient demographics.

ID	Sex	Age (Years)	Follow-Up Duration (Years)	Visual Acuity at Baseline (LogMAR)	Visual Acuity at Last Visit (LogMAR)	Genotype	OCT Grade at Last Examinations	Foveal Hypoplasia on OCT	ERG Rod Responses	ERG Cone Responses	Color Vision Test
#1	M	0.6	14.4	1.3/1.0	1.0/1.0	*CNGA3*	1	No	Normal	NM	Color blindness
#2	F	1.2	6.2	1.4/1.4	0.82/1.0	*CNGA3*	1	Yes	Not conducted	Not conducted	Not conducted
#3	M	1.5	10.6	0.7/0.82	0.7/0.7	*CNGA3*	2	No	Present	NM	Color blindness
#4	M	3.4	13.8	0.52/0.52	0.82/1.0	*CNGA3*	2	Yes	Decreased	NM	Not conducted
#5	M	2.5	6.2	0.7/1.0	1.0/1.0	*CNGA3*	1	No	Present	NM	Color blindness
#6	M	8	4.4	1.3/1.3	1.1/1.3	*CNGA3*	1	Yes	Present	NM	Not conducted
#7	M	12.4	0	0.7/0.7	0.7/0.7	*CNGA3*	1	No	Not conducted	Not conducted	Not conducted
#8	F	30.7	1	0.82/0.82	0.82/0.82	*CNGA3*	2	No	Not conducted	Not conducted	Color blindness
#9	F	0.5	14.9	1.3/1.3	1.4/1.4	*CNGB3*	3	Yes	Present	NM	Not conducted
#10	F	2.6	0.5	1.22/1.0	1.22/1.0	*CNGB3*	Not conducted	Not conducted	Present	NM	Not conducted
#11	M	38.2	1.8	0.82/1.0	0.82/1.0	*CNGB3*	3	Yes	Decreased	NM	Not conducted
#12	M	0.6	8.5	1.0/1.0	0.82/0.82	*PDE6C*	2	Yes	Present	NM	Color blindness
#13	M	0.6	25	1.0/1.0	1.0/1.0	*PDE6C*	4	No	Decreased	NM	Not conducted
#14	M	4.3	3.3	1.0/1.3	1.1/1.0	*PDE6C*	2	No	Normal	NM	Color blindness
#15	F	12.3	0.4	0.82/0.82	0.82/0.82	*PDE6C*	2	No	Present	NM	Not conducted
#16	F	13	14.2	1.3/1.3	1.0/1.0	*PDE6C*	4	No	Decreased	NM	Color blindness
#17	F	11.3	14.7	1.0/0.82	1.0/1.0	*PDE6C*	3	No	Present	NM	Not conducted
#18	M	16.2	2	0.82/0.82	0.82/0.82	*PDE6C*	3	No	Present	NM	Color blindness
#19	M	47.2	0	Not conducted	Not conducted	*PDE6C*	4	No	Not conducted	Not conducted	Not conducted
#20	M	9.7	2.4	1.52/1.52	1.4/1.4	*GNAT2*	1	No	Present	NM	Color blindness
#21	F	11.9	10	1.0/1.0	1.0/0.82	*GNAT2*	1	No	Present	NM	N/A

OCT grade: 1, continuous inner segment ellipsoid (ISe); 2, ISe disruption; 3, hyporeflective zone in the ellipsoid zone; 4, outer retinal atrophy. Rod and cone responses in the ERG were assessed based on the data in Table 2. Abbreviations: LogMAR, logarithm of the minimum angle of resolution; OCT, optical coherence tomography; ERG, electroretinogram; NM, not measurable.

**Table 2 genes-14-00519-t002:** Electroretinogram data.

No.	DA 0.01 b-Wave Amplitude		LA 3.0				LA 30 Hz	
			a-Wave Amplitude		b-Wave Amplitude		N1-P1 Amplitude	
	RE	LE	RE	LE	RE	LE	RE	LE
#1	215	196	7.54	7.93	0.485	3.24	NM	NM
#3	38	36.5	NM	NM	NM	NM	NM	NM
#4	40.8	69.4	NM	NM	NM	NM	NM	NM
#5	46.6	64.5	NM	NM	NM	NM	NM	NM
#6	37	40.6	Not conducted	Not conducted	Not conducted	Not conducted	NM	NM
#9	33.3	28.8	NM	NM	NM	NM	NM	NM
#10	41.3	8.3	NM	NM	NM	NM	NM	NM
#11	44.9	62.7	NM	NM	NM	NM	NM	NM
#12	44.0	31.1	NM	NM	NM	NM	NM	NM
#13	67.1	93.5	7.0	3.0	NM	NM	NM	NM
#14	NM	NM	NM	NM	NM	NM	NM	NM
#15	19.4	19.5	NM	NM	NM	NM	NM	NM
#16	85.1	77.9	NM	NM	NM	NM	3.64	1.62
#17	183.9	181.7	NM	NM	NM	NM	NM	NM
#18	34.8	35.2	NM	NM	NM	NM	NM	NM
#20	7.9	31.1	NM	NM	NM	NM	NM	NM
#21	176.3	134.8	NM	NM	NM	NM	NM	NM

Abbreviations: DA, dark-adapted; LA, light-adapted; RE, right eye; LE, left eye; NM, not measurable.

**Table 3 genes-14-00519-t003:** Genetic profiles.

No.	Gene	Nucleotide Variation	Protein Variation	Zygosity	CADD PHRED Score (GRCh37-v1.6)	Allele Frequency (%) in gnomAD	Pathogenicity (ACMG Classification and Its Criteria)	Previously Reported
#1	*CNGA3*	c.2T>A c.1001C>T	p.(Met1?) p.(Ser334Phe)	heterozygous	22.8 26		P (PVS1, PM2, PP3) LP	Novel [20]
#2	*CNGA3*	c.829C>T c.1001C>T	p.(Arg277Cys) p.(Ser334Phe)	Compound heterozygous	31 26	25/251350	P LP	[21] [20]
#3	*CNGA3*	c.553C>G c.1190G>T	p.(Leu185Val) p.(Gly397Val)	Compound heterozygous	20.5 25.7		LP LP	[22] [23]
#4	*CNGA3*	c.553C>G c.848G>A	p.(Leu185Val) p.(Arg283Gln)	Heterozygous	20.5 28.8	1/251394 16/251352	LP P	[22] [21]
#5	*CNGA3*	c.1262del c.1642G>A	p.(Lys421Serfs*44) p.(Gly548Arg)	Compound heterozygous	32 28.1	6/251292	P P	[24] [25]
#6	*CNGA3*	c.553C>T c.847C>T	p.(Leu185Val) p.(Arg283Trp)	Heterozygous	20.5 26.5	1/251394 25/251318	LP LP	[22] [26]
#7	*CNGA3*	c.829C>T	p.(Arg277Cys)	Homozygous	31	24/251350	P	[21]
#8	*CNGA3*	c.158_161dup c.1190G>T	p.(Arg55Aspfs*6) p.(Gly397Val)	Heterozygous	24 25.7		LP (PVS1, PM2) LP	Novel [22,23]
#9	*CNGB3*	c.1928+2T>C		Homozygous	33		P	[24]
#10	*CNGB3*	c.1258_1277del c.1579-2A>G	p.(Ile420Phefs*35)	Compound heterozygous	33 33	1/249908	P (PVS1, PP1) P	Novel [27]
#11	*CNGB3*	c.1579-2A>G Exon 16 deletion		Heterozygous	33	1/249908	P LP (PVS1, PM2)	[27] Novel
#12	*PDE6C*	c.1771G>A c.2269C>T	p.(Glu591Lys) p.(Gln757*)	Compound heterozygous	28.1 50		LP LP (PVS1, PM2)	[28] Novel
#13	*PDE6C*	c.1646T>C c.1766A>G	p.(Met549Thr) p.(Asp589Gly)	Compound heterozygous	22.5 24.6	2/251386	LP US	[8] [29]
#14	*PDE6C*	c.85C>T c.712C>T	p.(Arg29Trp) p.(Arg238*)	Heterozygous	24.1 36	6/277228 2/246146	P P	[30] [31]
#15	*PDE6C*	c.480G>T Exon 1 deletion	p.(Lys160Asn)	Compound heterozygous	33	1/245444	P P (PVS1, PM2, PP3)	[32] Novel
#16	*PDE6C*	c.85C>T c.1771G>A	p.(Arg29Trp) p.(Glu591Lys)	Compound heterozygous	24.1 28.1	6/282886 2/251120	P LP	[30] [28]
#17	*PDE6C*	c.1643G>T c.2507delG	p.(Trp548Leu) p.(Gly836Glufs*21)	Heterozygous	31 26.1	2/251354	LP (PVS1, PP1) P (PS3, PM1, PM2, PM3, PP3)	Novel Novel
#18	*PDE6C*	c.85C>T c.827G>A	p.(Arg29Trp) p.(Arg276Gln)	Compound heterozygous	24.1 26.7	6/282886 5/282768	P LP	[30] [33]
#19	*PDE6C*	c.1540T>A c.1785G>A	p.(Phe514Ile) p.(Met595Ile)	Heterozygous	21.3 27.8		US (PM1, PM2, PM3) US (PM2, PM3, PP3)	Novel Novel
#20	*GNAT2*	c.481C>T	p.(Arg161*)	Homozygous	36	5/282860	P	[22]
#21	*GNAT2*	c.730_743del c.481C>T	p.(His244Serfs*7) p.(Arg161*)	Heterozygous	34 36	5/152144	LP (PVS1, PM2) P	Novel [22]

Abbreviations: P, pathogenic; LP, likely-pathogenic; US, Uncertain significance; ACMG, American College of Medical Genetics and Genomics.

**Table 4 genes-14-00519-t004:** Comparison of patients with achromatopsia who were grouped by the causative genes with variants.

Characteristics	*CNGA3*	*CNGB3*	*PDE6C*	*GNAT2*	*p*-Value (PDE6C vs. CNGA3)
n, (%)	8 (38.1)	3 (14.3)	8 (38.1)	2 (9.5)	N/A
Age	7.6 ± 10.2 (0.6–30.7)	13.8 ± 21.2 (0.5–38.2)	13.2 ± 15.0 (0.6–47.2)	10.8 ± 1.6 (9.7–11.9)	0.430 ^a^
BCVA (LogMAR)					
Right	0.93 ± 0.35 (0.52–1.40)	1.12 ± 0.26 (0.82–1.30)	1.00 ± 0.16 (0.82–1.30)	1.26 ± 0.37 (1.00–1.52)	0.443 ^a^
Left	0.95 ± 0.29 (0.52–1.40)	1.10 ± 0.17 (1.00–1.30)	1.01 ± 0.21 (0.82–1.30)	1.26 ± 0.37 (1.00–1.52)	0.676 ^a^
Fundus					0.535 ^b^
Normal	6	3	5	2	
RPE change	1	0	0	0	
Retina/RPE atrophy	0	0	2	0	
ERG					
Rod response	Normal/subnormal				
Cone response	Absent				
Foveal hypoplasia					0.386 ^b^
Yes	3	2	1	0	
No	5	0	6	2	
OCT grade (worse eye)					0.011 (1 vs. 2,3,4) ^b^
1	5	0	0	2	
2	3	0	3	0	
3	0	2	3	0	
4	0	0	2	0	

^a^ Mann–Whitney U test, ^b^ Boschloo’s exact test. Abbreviations: LogMAR, logarithm of the minimum angle of resolution; RPE, retinal pigment epithelium; ERG, electroretinogram; OCT, optical coherence tomography.

**Table 5 genes-14-00519-t005:** Profiles of patients with achromatopsia worldwide.

Nation	Number of Patients, n	Mutated Genes, (%)				Remarks	Reference
		*CNGA3*	*CNGB3*	*PDE6C*	*GNAT2*		
Korea	19	38.1	14.3	38.1	9.5	Longitudinal analysis	This study
China	119	81.5	5.9	10.1	0.8	Identification of variants in *ATF6* (1.7%)	[7]
Netherland	63	4.8	87	N/A	0	High proportion of *CNGB3*:p.T383IfsX13 (80%)	[35]
UK/US	40	45	37.5	2.5	10	Long term follow-up	[1]
Italia	18	38.9	27.8	5.6	16.7	High proportion of homozygotes (61.9%)	[4]

## Data Availability

The data presented in this study are available on request from the corresponding author. The data are not publicly available due to privacy.

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
