# Peer review of "Clinical and Genetic Features of Korean Patients with Achromatopsia"

_genes, 2023, doi:10.3390/genes14020519_

Round 1

Reviewer 1 Report

Young Je Choi et al. analyzed 21 cases with achromatopsia in Korean population. More precisely, they looked at genotypes and phenotypes and did multiple phenotype-genotypes correlations. This is the first study to report such findings in Korea.

While the study is generally of interest and well presented. There are a few major points and multiple minor points to correct.

Major points:

Lines 1-11 (page 7): It is suprising to see significant p-values for small changes. The data are paired (baseline and follow up). A paired test such as paired samples Wilcoxon should be used instead of Mann-Whitney U test.

Line 38 (page 8): "two variants were frameshift (7.1%)" It should be 2/14 so 14.3% not 7.1%. There are many mistakes and english level is not sufficient in the whole paragraph (3.4 Genotypes). Please review it completely. For example:

Line 31 (page 8): "in four common" and not "of four common".

Line 33 (page 8): Change to "Eighteen patients were compound heterozygotes".

Line 34 (page 8): Change to "Sixteen of these variants".

Line 37 (page 8): "were" not "was"

Line 34-37 (page 8): 16 variants are novel. 10 are LP or P. 2 are VUS. What about the four other variants?

Criteria used for ACMG evalutation of novel variants should be provided.

The authors state that "Segregation was performed when possible." but compound heterozygous variants are reported in 18 of the 21 cases analyszed. To call variants coumpound heterozygous, segregation must be done. Was it done for these 18 cases?

Multiple variants reported as novel are in ClinVar database and therefore not novel:

CNGA3 c.848G>A

https://www.ncbi.nlm.nih.gov/clinvar/variation/9475/?new_evidence=true

PDE6C c.1766A>G

https://www.ncbi.nlm.nih.gov/clinvar/variation/970700/?new_evidence=true

PDE6C c.480G>T

https://www.ncbi.nlm.nih.gov/clinvar/variation/1213877/?new_evidence=true

PDE6C c.827G>A

https://www.ncbi.nlm.nih.gov/clinvar/variation/1017583/?new_evidence=true

GNAT2 c.481C>T

https://www.ncbi.nlm.nih.gov/clinvar/variation/522772/?new_evidence=true

Even given once as novel in patient #21 and reported in patient #20

Minor points:

Line 65: "ERG 1)" and not "ERG (1)"

Line 75: The reference 13 is not related to hypoplasia. Please cite a relevant publication.

Line 84: Please rephrase the sentence "A custom capture panel of 254 known genes and candidate genes was linked to 84 an inherited retinal disease".

Line 6 (page 7): Please add a space in "Figure1B".

Authors should be consistent reporting p-values with capital P or lowercase p in the whole manuscript.

Table 2: Header could be better done with RE and LE written above columns of numbers.

Table 3: How was the CNGB3 exon 16 deletion detected? Using which software? It is not described in the Methods. Are the breakpoints identified? Same comment for PDE6C exon 1 deletion.

Lines 72-73 (page 11): the difference in age is nearly significant and patients with normal ISe are on average 9.5 years younger. The authors should comment on that.

Table 5: Please remove the study from Tunisia. Only probands should be used to find percentage of cases per gene. Taking 12 individuals from the same family is wrong. They will have forcibly the same gene.

Author Response

We appreciate your comments.

Reviewer 2 Report

In the manuscript 'Clinical and Genetic Features of Korean Patients with Achromatopsia' by Choi et al., the authors aimed to characterize and correlate the genotype and phenotype of Korean patients with achromatopsia. The work's main strength is that they characterized Korean achromatopsia patients for the first time. Moreover, they found that in Korean patients, the retinal phenotypes of PDE6C-related achromatopsia variants were more likely to be worse than those of other genes. Another interesting finding is that the PDE6C achromatopsia variant has the highest occurrence compared to all other reported nationalities/ethnicities. These results are interesting and relevant to the field. They might be useful in future gene therapy strategies for achromatopsia patients.

Overall, the manuscript is scientifically sound; however, some points need to be addressed, specifically:

1. Introduction could be slightly expanded with more background description.

2. Interestingly, authors found decreased ERG rod responses; however, it requires more evidence or clarification.

In table 1, you indicate 'decreased' ERG rod responses in patients 4, 10, 11, 13, 15, 16, and 17. But then, in Table 2, you show scotopic ERG data only for b-wave amplitude, and not for a-wave, which describes rod responses. Therefore Table 2 should include a-wave amplitudes data, and under Table 1 or in the text, it should be stated that the conclusion 'decreased' is based on the data in Table 2.

Or, alternatively, describe in the result section what is this 'decrease' based on and show the data in a supplementary table or figure.

3. In Figure 2, OCT images, readable scales are missing

4. Page 1, line 17. It is unclear what exactly is significantly lower and what it refers to, 100% or 58.3%. Please re-write for clarity.

Author Response

We appreciate your comments.

Round 2

Reviewer 1 Report

The authors successfully answered my concerns.

I have only one last remark:

Criteria PM2 can be used for variants 8a, 12b and 21a to upgrade them from VUS to LP. Indeed they are at extremely low frequency for a recessive disorder.

Author Response

We appreciate your comment.
Please see the attachment.
